# Epidermal Growth Factor Receptor neddylation is regulated by a desmosomal-COP9 (Constitutive Photomorphogenesis 9) signalosome complex

Nicole Ann Najor[1,2], Gillian Nicole Fitz[2], Jennifer Leigh Koetsier[2], Lisa Marie Godsel[2,3], Lauren Veronica Albrecht[2], Robert Harmon[2], Kathleen Janee Green[2,3]*

[1]Department of Biology, College of Engineering and Science, University of Detroit Mercy, Detroit, United States; [2]Department of Pathology, Feinberg School of Medicine, Northwestern University, Chicago, United States; [3]Department of Dermatology Chicago, Feinberg School of Medicine, Northwestern University, Evanston, United States

**Abstract** Cell junctions are scaffolds that integrate mechanical and chemical signaling. We previously showed that a desmosomal cadherin promotes keratinocyte differentiation in an adhesion-independent manner by dampening Epidermal Growth Factor Receptor (EGFR) activity. Here we identify a potential mechanism by which desmosomes assist the de-neddylating COP9 signalosome (CSN) in attenuating EGFR through an association between the Cops3 subunit of the CSN and desmosomal components, Desmoglein1 (Dsg1) and Desmoplakin (Dp), to promote epidermal differentiation. Silencing CSN or desmosome components shifts the balance of EGFR modifications from ubiquitination to neddylation, inhibiting EGFR dynamics in response to an acute ligand stimulus. A reciprocal relationship between loss of Dsg1 and neddylated EGFR was observed in a carcinoma model, consistent with a role in sustaining EGFR activity during tumor progression. Identification of this previously unrecognized function of the CSN in regulating EGFR neddylation has broad-reaching implications for understanding how homeostasis is achieved in regenerating epithelia.

DOI: https://doi.org/10.7554/eLife.22599.001

*For correspondence:
kgreen@northwestern.edu

Competing interests: The authors declare that no competing interests exist.

## Introduction

Tissue morphogenesis and homeostasis are critically dependent on the proper spatial control of receptor tyrosine kinase activity (*Casaletto and McClatchey, 2012*). This is particularly so in complex regenerating tissues, such as the epidermis. The epidermis is a multi-layered epithelium that continuously renews itself through a highly choreographed program of morphological and biochemical changes, which are necessary to form a barrier against environmental insults and water loss (*Koster and Roop, 2007*). This process requires proliferating basal cells to stop dividing and undergo a commitment to differentiate as they transit toward the outer surface of the skin. Chemical cues governed by Epidermal Growth Factor Receptor (EGFR) and other signaling receptors must be tightly regulated to maintain a proper balance of proliferation and differentiation and to drive critical steps at each stage of epidermal stratification (*Fuchs and Nowak, 2008*; *Fuchs and Raghavan, 2002*; *Koster and Roop, 2007*; *Lopez-Pajares et al., 2013*).

**eLife digest** The outer layer of skin – the epidermis – forms a critical barrier against a range of stresses from the environment. The epidermis itself consists of multiple layers of cells that are constantly being renewed. New cells are made in the deepest layer and move upwards until they eventually reach the skin's surface. During this journey, the cells change the molecules they make in a process called epidermal differentiation.

To maintain an effective barrier, the epidermis must balance the division of cells in the deepest layer with the differentiation of cells in the layers above. When activated, a protein called the Epidermal Growth Factor Receptor (or EGFR for short) encourages cells in the deepest layer to divide. However, it remains poorly understood how the balance between cells dividing and cells differentiating is achieved.

The desmosome is a structure that can link together cells within the epidermis. Najor et al. now report a new interaction between the desmosome and a very large protein complex called the COP9- signalosome known to remove protein-based tags from other proteins. The experiments show that the COP9-signalosome results in the removal of these tags from EGFR. The status of the tags on EGFR regulates whether or not it is found at the cell surface. Najor et al. propose that that the desmosome acts as a scaffold and holds the COP9 signalosome close to EGFR. The enzyme in the COP9 signalosome then removes protein-based tags from EGFR, which triggers a series of events that remove EGFR from the cell surface. This dampens down the signals EGFR would normally send to make cells divide, and allows differentiation to proceed.

The balance between cell division and differentiation is a fundamental process that is affected in many skin conditions, including psoriasis and atopic dermatitis. EGFR is also commonly overactive in cancers. As such, understanding how epidermal differentiation and cell division are controlled will shed light on a variety of disorders, allowing for the potential development of new treatments for these diseases.

DOI: https://doi.org/10.7554/eLife.22599.002

While the mechanisms that dictate EGFR activity and dynamics during epidermal differentiation are not well understood, properly positioning the receptor in proximity to effector and regulatory machinery is likely to play a role. Candidate cytoarchitectural components that could serve in this manner are cell-cell junctions. In the epidermis, the most prominent of these junctions are desmosomes, which confer mechanical strength to tissues by linking the intermediate filament (IF) cytoskeleton through the IF anchoring protein Desmoplakin (Dp) to sites of desmosomal cadherin-mediated adhesion (*Kowalczyk and Green, 2013*). The importance of the desmosome-IF complex is underscored by the existence of mutations in genes encoding desmosome components that lead to skin, heart, and hair defects (*Lai-Cheong et al., 2007*; *Petrof et al., 2012*). While the desmosome has classically been regarded as an adhesion complex, recent studies have demonstrated that it can act as a signaling scaffold to ensure the spatial and temporal execution of the epidermal differentiation program (*Broussard et al., 2015*; *Harmon and Green, 2013*; *Schmidt and Koch, 2007*; *Sumigray and Lechler, 2015*).

In particular, we showed that the desmosomal cadherin, Desmoglein 1 (Dsg1), is not only required for maintaining epidermal tissue integrity in superficial layers, but also promotes keratinocyte differentiation as cells transit out of the basal layer (*Getsios et al., 2009*). It performs this function at least in part by attenuating the activity of ErB receptors (EGFR and ErbB2) and downstream MAPK signaling (*Getsios et al., 2009*; *Harmon et al., 2013*). Dsg1 is optimally positioned to exert this sort of spatial control on the EGFR pathway, as it is first expressed as cells commit to differentiation and becomes progressively concentrated in the superficial epidermal layers. Thus, it is a good candidate to serve as a scaffold for machinery necessary to down-regulate EGFR family activity, which is required for this commitment to differentiation. The unusually long cytoplasmic tail of Dsg1, but not the adhesive ectodomain, was required to suppress EGFR and Extra Cellular Signal-regulated Kinase 1/2 (Erk1/2) signaling, indicating a previously unrecognized function for Dsg1 that transcends its canonical roles in adhesion. However, the molecular machinery that integrates the desmosome-IF scaffold with the biochemical differentiation program is not well-defined.

To identify mediators of Dsg1 signaling functions we carried out a yeast-two-hybrid (Y2H) screen using the cytoplasmic domain of Dsg1 as bait (*Harmon et al., 2013*). Among the putative binding partners identified was the third subunit (Cops3) of the eight-subunit complex called the COP9 signalosome (CSN). Discovered in 1994 in *Arabidopsis*, the Constitutively Photomorphogenic (COP) mutants, *cops1-8,* exhibit a photomorphogenic phenotype in complete darkness (*Chory et al., 1989*; *Deng et al., 2000*; *Denti et al., 2006*; *Miséra et al., 1994*; *Wei et al., 1994*; *Wei and Deng, 1996*; *Wei and Deng, 1999*). The CSN functions as a Nedd8 isopeptidase to remove Nedd8 moieties from its substrates, an activity depending on the Cops5 subunit. This hydrolysis has been classically studied in the context of de-neddylation of cullin-RING (Really Interesting New Gene) E3 ligases (Cul1, 2, 3, 4a, 4b, 5, 7) (*Cope et al., 2002*; *Duda et al., 2008*; *Saha and Deshaies, 2008*), which regulate fundamental processes important for development and tissue homeostasis, including cell cycle, signal transduction, transcription and DNA replication (*Petroski and Deshaies, 2005*). To promote the activity of cullin-RING E3 ligases, the CSN de-neddylates and partners with de-ubiquitinase enzymes to 'de-ubiquitinate' and rescue the cullin from spurious self-ubiquitination that occurs when the Nedd8 modification is present (*Hetfeld et al., 2005*; *Wee et al., 2005*; *Zhou et al., 2003*). These functions create activation cycles allowing cullin-RING E3 ligases to focus their activity on specific substrates. Cells deficient for the CSN are unable to remove Nedd8, and as a consequence cullin-RING E3 ligase subunits are autoubiquitinated resulting in reduced function (*Bosu and Kipreos, 2008*; *Hetfeld et al., 2005*; *Hotton and Callis, 2008*; *Schwechheimer et al., 2001*; *Wee et al., 2005*; *Wu et al., 2005*; *Zhou et al., 2003*). It has yet to be demonstrated whether the CSN is able to de-neddylate other substrates modified by Nedd8.

In this report, we show that the Cops3 subunit of the CSN interacts with desmosomal components, Dsg1 and Dp. Loss of Cops3 results in an increase in phosphorylated EGFR (pEGFR), which is associated with compromised keratinocyte differentiation, suggesting that Cops3, and consequently the CSN, inhibits EGFR signaling to promote differentiation. Further, Cops3 deficiency compromises the ability of ectopic Dsg1 to promote expression of keratinocyte differentiation markers, consistent with functional cooperation between desmosomes and the CSN. We go on to show that EGFR is neddylated in human keratinocytes and genetically interfering with desmosomal components or Cops3 results in elevated EGFR receptor neddylation, suggesting that EGFR is a non-cullin substrate for CSN-dependent de-neddylation. Importantly, this increase in neddylation was accompanied by a decrease in ubiquitination and altered EGFR dynamics in response to an acute ligand stimulus. A reciprocal relationship between Dsg1 and neddylated EGFR was observed in a 3D carcinoma model, raising the possibility that loss of desmosomes during cancer progression unscaffolds membrane-associated CSN complexes, resulting in hyper-neddylated EGFR. These data support a model whereby epidermal differentiation is assisted by desmosome-dependent scaffolding of the CSN complex to down-regulate EGFR through removal of Nedd8 modifications.

## Results

### The COP9 signalosome (CSN) interacts with the desmosome

We recently identified a role for the desmosomal cadherin, Dsg1, in promoting epidermal differentiation through attenuation of EGFR/MAPK signaling (*Getsios et al., 2009*). This function did not require the Dsg1 adhesive ectodomain, but did require the unique, extended cytoplasmic tail. While a Dsg1 binding partner, Erbin, was shown to interfere with Ras-Raf coupling downstream of EGFR (*Harmon et al., 2013*), the upstream regulation of EGFR is still poorly understood. A Y2H screen using the Dsg1 tail as bait revealed the CSN subunit, Cops3, as a new Dsg1 binding partner. Cops3 could interact with the constructs comprising the full cytoplasmic tail of Dsg1 or a construct lacking only the membrane proximal intracellular anchor (IA) in yeast (*Figure 1a*). The interaction with Cops3 was selective, as the fourth subunit of the CSN was unable to interact with the Dsg1 tail in yeast (*Figure 1b*). To validate the interaction, we tested whether recombinant Dsg1 (GST-Dsg1) can associate with endogenous Cops3 in normal human keratinocyte (NHEK) lysates *in vitro*. Indeed, purified GST-Dsg1 bound to sepharose beads specifically precipitated endogenous Cops3 from NHEK lysates (*Figure 1c*).

To address whether Cops3 engages more broadly with the desmosome, we carried out immuno-precipitations of Dp, an IF-anchoring protein present in all desmosomes. Cops3 was also present in

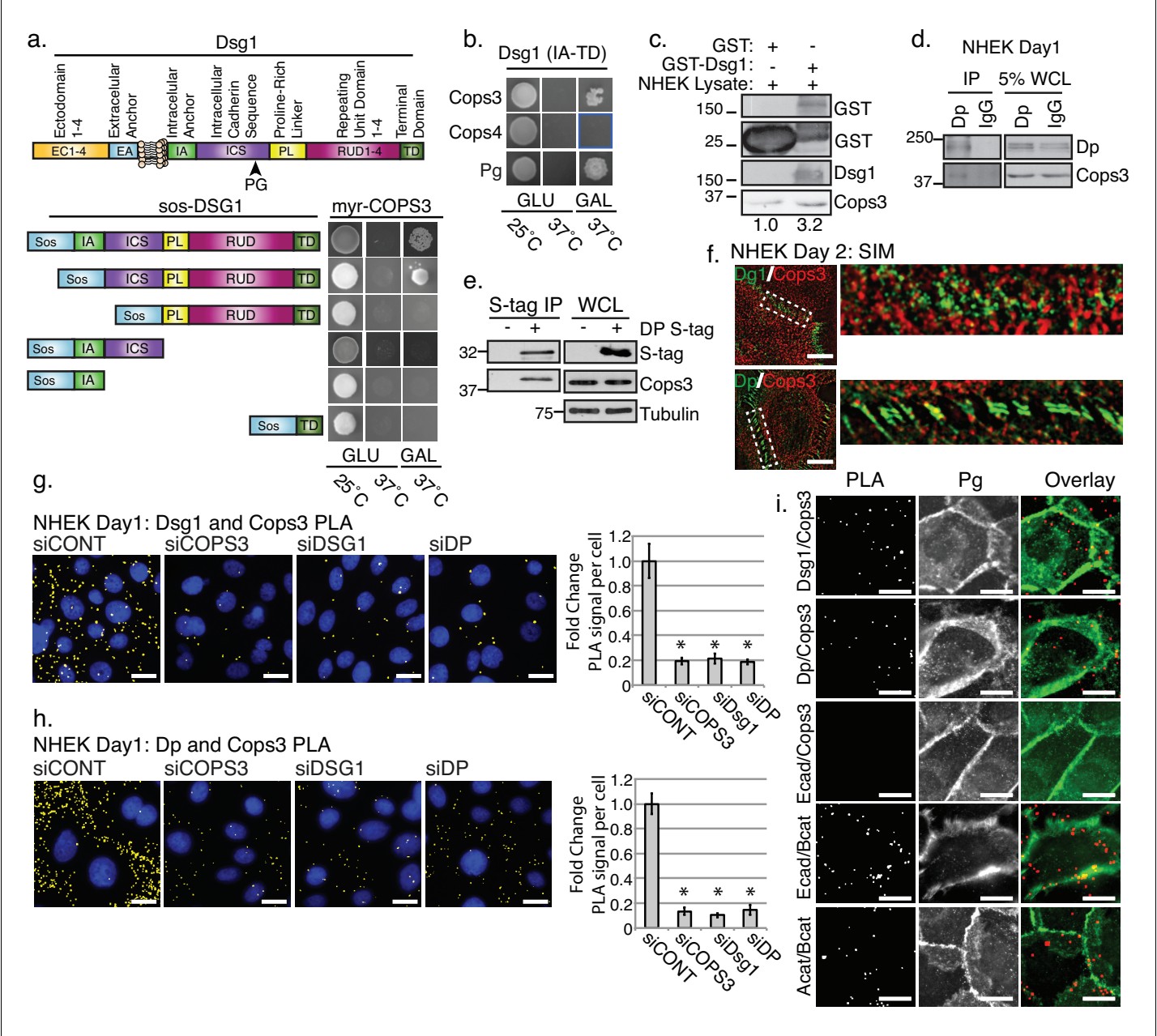

**Figure 1.** The third subunit of the COP9 signalosome (CSN), Cops3, interacts with desmosomal components Desmoglein1 (Dsg1) and Desmoplakin (Dp). (a) Cytotrap yeast two hybrid (Y2H) indicating an interaction between Cops3 and the cytoplasmic tail of Dsg1. Growth at 25°C is permissive; growth on galactose (Gal) at 37°C confirms interaction. (b) Cops4 of the CSN does not interact with Dsg1. A known binding partner of Dsg1, Plakoglobin (Pg), was used as a positive control. (c) Purified GST-Dsg1 (includes the entire cytoplasmic domain), was bound to sepharose beads and incubated with normal human keratinocyte (NHEK) lysates. (d) An endogenous Dp immunoprecipitation (IP) was performed in NHEKs after 1 day of differentiation, indicating an interaction with Cops3. (e) A 32 kDa S-tagged C-terminal truncation of Dp (DP-S-tag) associates with Cops3 in squamous cell carcinoma line 9 (SCC9) lysates. (f) Structured Illumination Microscopy (SIM) of Cops3 (red) and Dsg1 or Dp (green) in NHEKs after 2 days of differentiation, shows close proximity between Cops3 and desmosome components at junctions. (g and h) Proximity ligation assay (PLA) in NHEK cultures with siRNA-targeting COPS3, DSG1, DP, or scramble sequence (siCONT) after 1 day of differentiation. Quantification is displayed to the right of the image panels. Cops3/Dsg1= *p<0.001, and Cops3/Dp= *p<0.00005, Student's t test; mean ±SEM. (i) PLA in NHEK cultures after 1 day of differentiation with adherens junction components (Ecadherin, E-cad, Beta catenin, B-cat, Alpha catenin, A-cat) and co-stained with Plakoglobin to mark cell borders. Proximal proteins within 40–100 nm appear as red but have been pseudo-colored yellow in (g) and (h) and have been dilated in (i) for ease of visualization in Fiji. All experiments are representative of 3 or more independent repeats. Scale bars = 20 μm.

DOI: https://doi.org/10.7554/eLife.22599.003

The following source data and figure supplements are available for figure 1:

*Figure 1 continued on next page*

*Figure 1 continued*

**Source data 1.** PLA analysis of Dsg1 and Cops3 in NHEKs differentiated for 1 day.
DOI: https://doi.org/10.7554/eLife.22599.008
**Source data 2.** PLA analysis of Dp and Cops3 in NHEKs differentiated for 1 day.
DOI: https://doi.org/10.7554/eLife.22599.009
**Source data 3.** PLA analysis of desmosomal and adherens junctions components in NHEKs differentiated for 1 day.
DOI: https://doi.org/10.7554/eLife.22599.010
**Figure supplement 1.** Cytotrap Y2H indicating no interaction between Cops3 and the N-terminal portion of Dp (amino acids 1–584).
DOI: https://doi.org/10.7554/eLife.22599.004
**Figure supplement 2.** Western blots indicating knockdown efficiency of NHEKs used in proximity ligation assay (PLA) of Dsg1 and Cops3.
DOI: https://doi.org/10.7554/eLife.22599.005
**Figure supplement 3.** Western blots indicating knockdown efficiency of NHEKs used in proximity ligation assay (PLA) of Dp and Cops3.
DOI: https://doi.org/10.7554/eLife.22599.006
**Figure supplement 4.** Quantification of PLA analysis in NHEKS after 1 day of differentiation.
DOI: https://doi.org/10.7554/eLife.22599.007

immunoprecipitates of endogenous Dp (*Figure 1d*). To narrow down the domains in Dp that were necessary for Cops3 interaction we performed a Y2H analysis utilizing Dp truncation constructs. While the N-terminal portion of Dp harboring the amino acid residues 1–584 (DP-NTP) was not sufficient for Dp's interaction with Cops3 (*Figure 1—figure supplement 1*), a 32 kDa truncated version of the C-terminal tail of Dp co-precipitated with endogenous Cops3 in the SCC9 cell line, which was used to overcome the limitations of ectopic expression of the Dp S-tag fusion protein (DP-S-tag) in primary cells. This suggested the site of interaction on Dp resides within the C-terminus (*Figure 1e*). To further examine the relative disposition of Cops3, Dsg1 and Dp we carried out structured illumination microscopy (SIM), which produces two times the resolution of conventional optical microscopes, resulting in the minimal overlap of associated proteins in co-localization. SIM images revealed Cops3 to be distributed throughout the cell, impinging upon Dsg1 or Dp concentrated at cell-cell junctions. Cops3 was also closely associated with cytoplasmic Dsg1 and Dp particles, which could represent vesicles or cytoplasmic precursors, previously described (*Godsel et al., 2005*) (*Figure 1f*).

To further address the interaction between desmosomes and the CSN we took advantage of the proximity ligation assay (PLA), a powerful technique that provides an indication of protein proximity from 40 to 100 nm to examine the spatial relationship of endogenous Dsg1 and Cops3 in cells (*Weibrecht et al., 2010*). A positive PLA signal was observed for the Dsg1 and Cops3 antibody pair, which was abrogated by treatment with either *DSG1* or *COPS3* siRNA (*Figure 1g* and *Figure 1—figure supplement 2*), consistent with an association between Dsg1 and Cops3 in NHEKs. In addition, PLA analysis revealed a positive signal for Dp and Cops3 in NHEKs treated with high calcium media to induce epidermal differentiation, which was likewise inhibited through knockdown of Dp and Cops3 (*Figure 1h* and *Figure 1—figure supplement 3*). To investigate further the possibility that the entire desmosome serves as a docking site for the recruitment of the CSN, we also assessed whether Dp knockdown diminished Cops3's ability to co-localize with Dsg1 (*Figure 1g*) and vice versa (*Figure 1h*). In both cases, PLA signals were significantly reduced, supporting the idea that the integrity of the entire desmosome is required to scaffold the COP9 signalosome.

To address the specificity of Cops3 interactions with desmosomes we paired the Cops3 antibody with the adherens junction protein E-cadherin (E-cad). As a positive control, E-cad and Beta-catenin (B-cat) were paired as well as Alpha-catenin (A-cat) and B-cat (*Figure 1i* and *Figure 1—figure supplement 4*). These experiments clearly show that while Cops3 interacts with desmosomal molecules Dsg1 and Dp, it does not interact with adherens junction components. In addition, we used the cell-cell junction protein Plakoglobin (Pg) as a fluorescence marker for cell-cell junctions to determine the extent to which PLA signals co-localize with cell-cell interfaces. These data show that while not all desmosome-Cops3 interactions are in close proximity to the Pg signal, a large percentage of PLA dots do appear to be localized at cell-cell interfaces (~60% for Dsg1/Cops3;~67% for Dp/Cops3), although to a somewhat lesser extent than the positive control PLA pairings Alpha-Catenin/Beta-Catenin (~82%) and Ecad/Beta-catenin (~88%) (*Figure 1i* and *Figure 1—figure supplement 4c*). Cytoplasmic Cops3 interactions with desmosome molecules may occur between non-membrane

bound Dp precursors and endocytosed membrane associated Dsg1 vesicles that we have previously identified in these cells, which could account for cytoplasmic PLA signal (*Godsel et al., 2005*; *Harmon et al., 2013*; *Nekrasova et al., 2011*; *Patel et al., 2014*).

## Dsg1-induced differentiation requires Cops3

We previously showed that Dsg1 promotes epidermal differentiation and that the adhesive ectodomain is dispensable for this function (*Getsios et al., 2009*). Towards addressing whether the interaction between Cops3 and the desmosome contribute to this function of the Dsg1 cytoplasmic domain, we silenced *COPS3* in NHEKs through electroporation of siRNAs targeted specifically against this CSN subunit. Biochemical analysis revealed a reduction in differentiation related gene products including Dsg1, Desmocollin1 (Dsc1), Keratin1 (K1), and Keratin 10 (K10) (*Figure 2a*). Similar impairment of these features of the keratinocyte differentiation program were observed using five oligos each with independent targets within the *COPS3* sequence (siCOPS3-465, siCOPS3-Dharma3, siCOPS3-IDT pool including 3 oligos), supporting the idea that the observed changes were not due to off target effects (*Figure 2—figure supplement 1*). To further address whether Cops3 contributes to proper epidermal differentiation and morphogenesis, we generated Cops3 deficient 3D organotypic cultures lifted to an air-medium interface (epidermal raft cultures) and harvested 3 days later (*Figure 2b*). Biochemical analysis of raft lysates revealed similar differentiation defects when compared with monolayer cultures reflected by a decrease in Dsg1, Dsc1, the suprabasal keratins K1 and K10 and the cell envelope protein Loricrin (Lor) (*Figure 2c*). Immunofluorescence of siCOPS3 Day 3 rafts also revealed defects in the differentiation marker K10 and Dsg1 (*Figure 2d*). Interestingly, we noted a decrease in Dsg1 protein expression (*Figure 2a and c*), which could contribute to the differentiation defect (*Getsios et al., 2009*). To test whether aberrant differentiation was due to the loss of Dsg1 protein expression, we introduced ectopic Dsg1-FLAG in the background of NHEKs deficient for Cops3. As we previously reported, ectopic Dsg1 promotes expression of differentiation specific markers in control cultures (*Getsios et al., 2009*). Biochemical analysis of cells retrovirally transduced with a Dsg1-Flag vector revealed that Dsg1-Flag was unable to drive expression of Dsc1, K1, and K10 in Cops3-silenced cultures after inducing differentiation for 2 days to the same extent as control cells (*Figure 2e*). To directly capture the impact of siCOPS3 on early differentiation in a specific population of cells expressing Dsg1, we scored NHEKs transiently expressing a GFP-tagged version of Dsg1 for an early differentiation marker, K10, in control and *COPS3*-silenced cultures after 1 day of differentiation. We observed that Dsg1-GFP expressing cells were unable to express K10 upon silencing of *COPS3* (*Figure 2f and g*). Thus, Dsg1 requires Cops3 for efficient Dsg1-stimulated keratinocyte differentiation.

## Inhibition of EGFR signaling restores differentiation in Cops3-deficient keratinocytes

Since the loss of Cops3 interferes with differentiation, we addressed the extent to which EGFR and downstream effectors, Erk and Mek, are affected upon the silencing of *COPS3*. Increases in pEGFR, phosphorylated Erk (pErk), and phosphorylated Mitogen Activated Protein Kinase (pMek) were observed (*Figure 3a*). These increases in MAPK signaling corresponded to a decrease in differentiation markers Dsg1, Dsc1, K10 and Lor (*Figure 3a* right panel and *Figure 2a*), consistent with our previous demonstration that the canonical Ras-Raf-Erk pathway is functionally linked to Dsg1's ability to promote differentiation (*Getsios et al., 2009*; *Harmon et al., 2013*). An increase in pErk was also observed in 3D organotypic raft cultures treated with siRNA targeting *COPS3* (*Figure 3b*). To test whether inhibition of EGFR is capable of restoring differentiation in Cops3-deficient cultures, we treated NHEK siRNA-targeted *COPS3* cells with the specific EGFR inhibitor AG1478. Pharmacological inhibition of EGFR restored Dsg1, Dsc1, and K10 expression without increasing Cops3 protein levels (*Figure 3c*). These data are consistent with the hypothesis that the CSN, recruited to the desmosome through Cops3, is required to properly modulate EGFR activity.

## EGFR neddylation is elevated in desmosome and Cops3 deficient cells

While cullins are the best-known substrates for the Nedd8 modification, EGFR neddylation was previously reported to occur in CHO (chinese hamster ovary) cells (*Oved et al., 2006*). More recently Transforming Growth Factor Beta Receptor Type II (TGFβRII) was shown to be regulated by

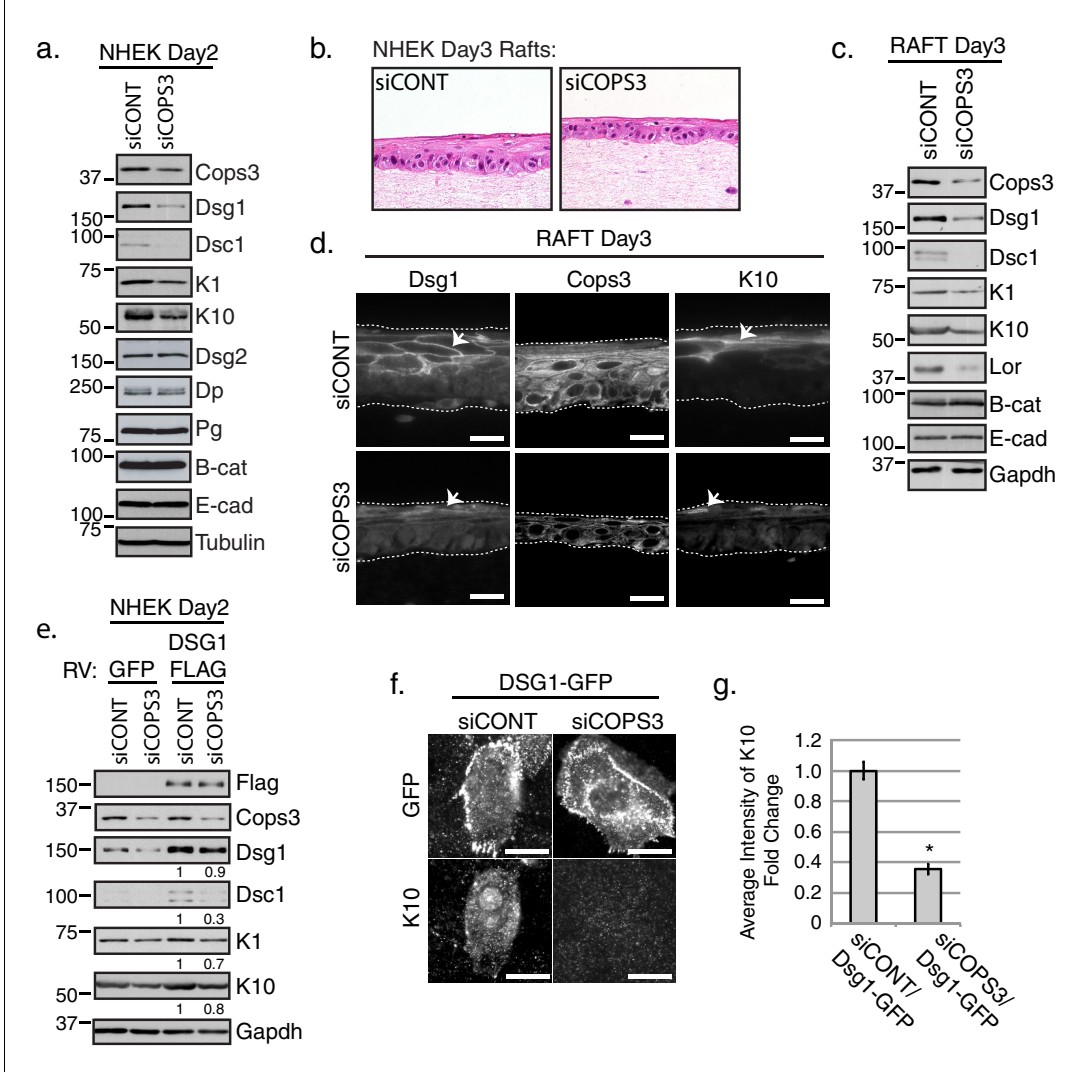

**Figure 2.** Keratinocyte differentiation requires Cops3. (a) RNAi-mediated silencing of COPS3 in NHEK cultures differentiated for 2 days in 1.2mM Ca[+][+] results in a decrease in differentiation markers: Dsg1, Desmocollin1 (Dsc1), Keratin 10 (K10) and Keratin1 (K1). No change in protein expression for Desmoglein2 (Dsg2), Dp, Pg, Beta-catenin (B-cat), Ecadherin (E-cad) was seen between siCONT and siCOPS3 cell lysates. Tubulin was utilized as a loading control. (b) 3D epidermal raft cultures were generated with cells silenced by siRNA-targeting COPS3 or scramble sequence (siCONT) and harvested 3 days after lifting to an air-medium interface. H & E staining and biochemical analysis of raft cultures display signs of impaired differentiation. (c) Biochemical analysis of 3D raft lysates corresponds to morphology changes with an observed decrease in Dsg1, Dsc1, K10, K1, and Loricrin (Lor) with no changes in B-cat and E-cad. (d) Immunofluorescence of siCOPS3 raft cultures displays a decrease in Dsg1 and K10 (scale bar = 20 μm). (e) Cells silenced for Cops3 were transduced with GFP or Dsg1-FLAG retrovirus (RV). Ectopic expression of Dsg1 was unable to fully compensate for the differentiation defect found in Cops3 deficient cells, which was analyzed by differentiation markers (Dsc1, K1, K10). Densitometry quantifications of differentiation markers are represented as fold change relative to Gapdh and indicated below the blot. (f) NHEKs transiently transfected with the Dsg1-GFP construct and processed after 1 day of differentiation do not express K10 upon the silencing of COPS3 (scale bar = 20 μm). (g) Quantification of average fluorescence intensity of K10 in Dsg1-GFP expressing cells. *p<0.00005 (Student's t test); mean ±SEM. All experiments are representative of 3 or more independent repeats.

DOI: https://doi.org/10.7554/eLife.22599.011

The following source data and figure supplement are available for figure 2:

**Source data 1.** Cells transiently transfected with Dsg1-GFP do not express K10 upon the silencing of COPS3.
DOI: https://doi.org/10.7554/eLife.22599.013

**Figure supplement 1.** Multiple siRNA targets of COPS3 displayed keratinocyte differentiation defect and an increase in EGFR signaling in keratinocytes differentiated for 1 day in 1.2 mM Ca[++].
DOI: https://doi.org/10.7554/eLife.22599.012

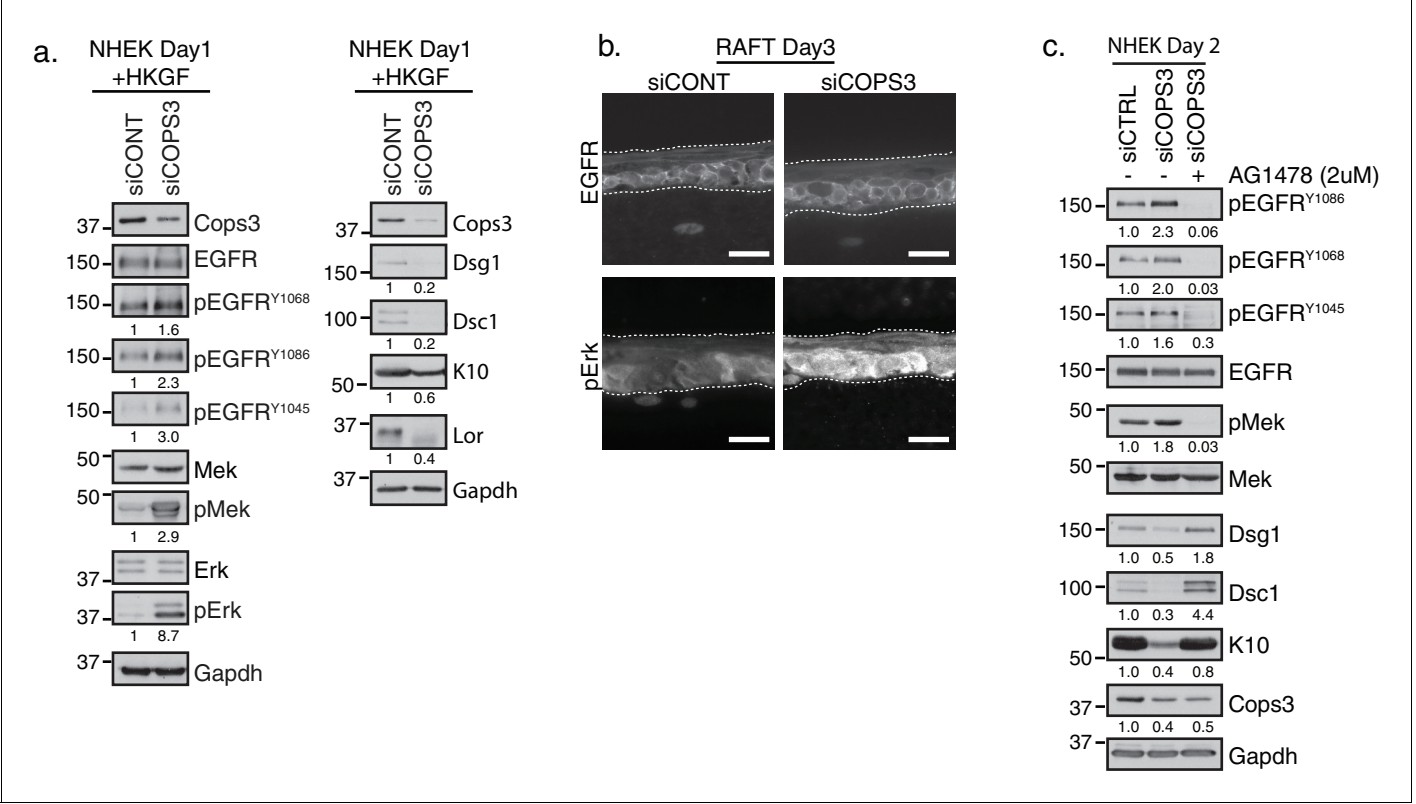

**Figure 3.** Silencing of Cops3 promotes Epidermal Growth Factor Receptor (EGFR) signaling in NHEKs. (a) RNAi-mediated silencing of COPS3 in NHEKs differentiated for 2 days in 1.2 mM Ca$^{++}$ with Human Keratinocyte Growth Factors (HKGF) display a decrease in differentiation markers (right panel) and an increase in phosphorylation of EGFR, along with its downstream effectors (Mek and Erk), indicating an overall increase in MAPK signaling (left panel). For quantification of phosphorylated proteins, each band was normalized to the amount of total protein then compared relative to the control culture (siCONT) and indicated below each western blot. (b) Immunofluorescence of siCOPS3 raft cultures display similar levels of total EGFR and an increase in pErk signal (scale bar = 20 μm). (c) Western blot analysis of NHEKs treated with siRNA targeting scramble (siCONT) and COPS3 (siCOPS3) and incubated with an EGFR specific inhibitor, AG1478, at 2 μM. EGFR inhibition restored the expression of the differentiation markers, Dsg1, Dsc1, and K10, indicating that Cops3 operates upstream of EGFR signaling in the epidermal differentiation program. Densitometry quantification is represented as fold change relative to Gapdh and indicated below the blot. All experiments are representative of 3 or more independent repeats.
DOI: https://doi.org/10.7554/eLife.22599.014

neddylation (*Zuo et al., 2013*), raising the possibility of receptor neddylation as a means of modulating EGFR's activity. To address whether EGFR neddylation occurs in NHEKs and whether the desmosome-CSN complex regulates EGFR neddylation status, we performed RNAi-mediated silencing of *DSG1*, *DP*, and *COPS3* in NHEKs differentiated for 1 day and assessed the association of the Nedd8 moiety with EGFR using PLA. A significant increase in PLA positive signals between EGFR and Nedd8 was observed when *DP*, *DSG1* or the CSN component *COPS3* was silenced (*Figure 4a* and *Figure 4—figure supplement 1*), supporting a model whereby both the desmosome and the CSN are necessary for regulating EGFR de-neddylation. To examine the levels of EGFR neddylation biochemically, we performed EGFR immunoprecipitations in SCC9 cells transiently expressing HA-tagged Nedd8, to facilitate the detection of the modification. In this case, the use of this cell line helped overcome limitations of ectopic expression in primary cells. SCC9 cells are devoid of Dsg1 when grown in a monolayer, however they do express other desmosomal cadherins, including desmoglein 2, and assemble robust desmosomes that include Dp (*Chen et al., 2012*; *Godsel et al., 2005*; *Nekrasova et al., 2011*). Loss of either Dp or Cops3 via RNAi-mediated silencing resulted in elevated EGFR neddylation (*Figure 4b*). The fact that *DP* knockdown in desmosome-containing SCC9 cells impacts EGFR neddylation status is consistent with the observation that loss of either Dsg1 or Dp interferes with recruitment of Cops3 to desmosomes in NHEK cells (*Figure 1g and h*).

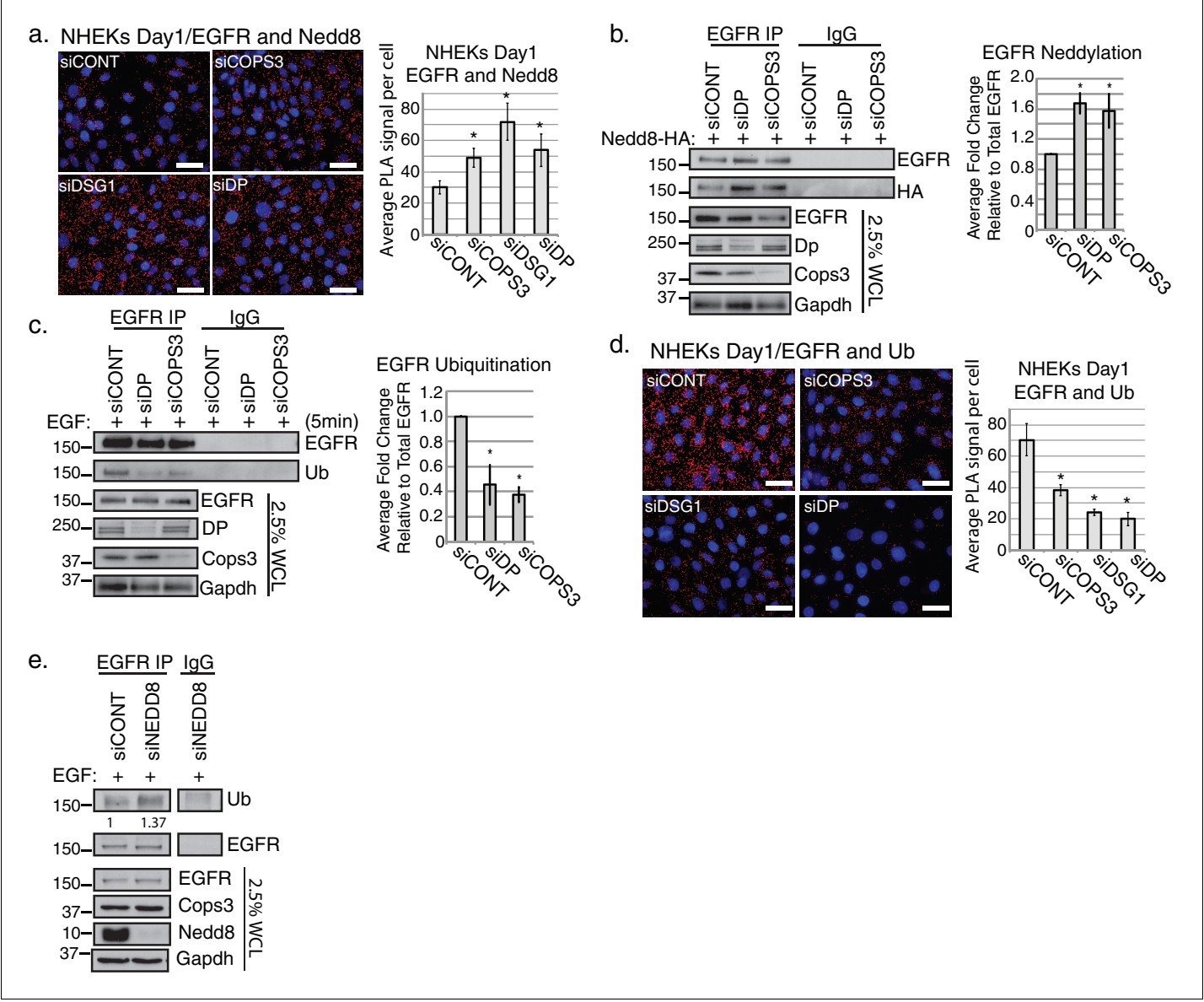

**Figure 4.** Loss of desmosomal components, Cops3 or Nedd8 affects the biochemical status of EGFR. (a) PLA analysis using Nedd8 and EGFR antibodies in NHEKs treated with siRNA targeting DP, DSG1, COPS3 or scramble (siCONT) (scale bar = 20 μm). Quantification of PLA signals displayed to the right of the images. *p<0.05 (Student's t test); mean ±SEM. (b) EGFR immunoprecipitations (IPs) in SCC9 cells harboring the Nedd8-HA construct indicate an increase in EGFR neddylation upon silencing of Cops3 or Dp. Densitometry quantification of 4 independent replicates is shown to the right of the representative blot. *p<0.05 (Student's t test); mean ±SEM. (c) Cells treated with siRNA directed towards COPS3 or DP display a decrease in EGFR ubiquitination. EGFR IPs in SCC9 cells treated with 50 ng/ml EGF for 5 min to induce internalization of the receptor. Densitometry quantification of 4 independent replicates is shown to the right of the representative blot. *p<0.01 (Student's t test); mean ±SEM. (d) PLA analysis using Ubiquitin and EGFR antibodies in NHEKs treated with siRNA targeting DP, DSG1, COPS3 or scramble (siCONT) (scale bar = 20 μm). Quantification of PLA signals displayed to the right of the images. *p<0.05 (Student's t test); mean ±SEM. (e) EGFR immunoprecipitations (IPs) in SCC9 cells indicate EGFR ubiquitination upon silencing of Nedd8 and treated with 50 ng/ml EGF ligand for 5 min. Western blot displayed is a representative blot of 3 independent repeats. All experiments are representative of 3 or more independent repeats.

DOI: https://doi.org/10.7554/eLife.22599.015

The following source data and figure supplements are available for figure 4:

**Source data 1.** PLA analysis of Nedd8/EGFR antibodies and Ubiqutin/EGFR antibodies in NHEKs.
DOI: https://doi.org/10.7554/eLife.22599.018

**Source data 2.** Analysis of Nedd8-EGFR and Ubiquitin-EGFR through EGFR immunoprecipitations (IPs) in SCC9 cells.
DOI: https://doi.org/10.7554/eLife.22599.019

*Figure 4 continued on next page*

*Figure 4 continued*

**Source data 3.** PLA analysis of EGFR/Nedd8 and EGFR/Ub upon the loss of Nedd8.
DOI: https://doi.org/10.7554/eLife.22599.020
**Figure supplement 1.** Western blots indicating knockdown efficiency of NHEKs used in proximity ligation assay (PLA) of EGFR/Nedd8 and EGFR/Ub (Fig.
DOI: https://doi.org/10.7554/eLife.22599.016
**Figure supplement 2.** EGFR is ubiquitinated upon loss of Nedd8 by PLA analysis in NHEKs differentiated for 1 day in high calcium.
DOI: https://doi.org/10.7554/eLife.22599.017

Since neddylation and ubiquitination occur on lysines, we questioned whether neddylation competes with ubiquitination on EGFR. To assess endogenous ubiquitination of EGFR we treated the cells with EGF (50 ng/ml) for 5 min, then performed EGFR immunoprecipitations in SCC9 cells. Upon loss of Cops3 and Dp we observed reduced ubiquitination of EGFR (*Figure 4c*). We also found a decrease in a PLA signal between EGFR and Ubiquitin when Dsg1, Cops3, or Dp was silenced (*Figure 4d*).

A previous report suggested that neddylation of ectopically expressed EGFR in CHO cells is required for subsequent ubiquitination and turnover of the receptor (*Oved et al., 2006*), whereas our observations are consistent with a reciprocal relationship between neddylation and ubiquitination. In order to directly address whether ubiquitination of endogenous EGFR depends on the Nedd8 modification human epithelial cells, EGFR ubiquitination was assayed in SCC9 cells silenced for *NEDD8*. Upon immunoprecipitation of EGFR, it was found that EGFR is ubiquitinated in Nedd8 depleted cells, on average about 1.4-fold higher than control RNAi-treated cells (*Figure 4e*). Additionally, PLA analysis revealed an association between EGFR and Ubiquitin upon the loss of Nedd8 (*Figure 4—figure supplement 2*). Overall, this suggests that both desmosome and CSN components support the generation and/or maintenance of ubiquitinated EGFR, at the expense of EGFR neddylation.

## The role of desmosomes and the CSN in EGFR dynamics

Ubiquitination is important for the internalization and post-internalization sorting of EGFR, where the ubiquitin ligase, Cbl, coordinates the transfer of the Ubiquitin from an E2 enzyme to the receptor's cytoplasmic domain (*Conte and Sigismund, 2016*; *Joazeiro et al., 1999*; *Levkowitz et al., 1999*; *Sorkin and Goh, 2008*; *Waterman et al., 1999*; *Yokouchi et al., 1999*). When EGFR fails to acquire Ubiquitin, it is recycled back to the membrane (*Levkowitz et al., 1998*). Further, neddylation has previously been shown to stabilize substrates including other membrane receptors such as TGFβRII (*Zuo et al., 2013*). Together with the results described above, this raises the possibility that by scaffolding the CSN near EGFR, desmosomes might promote EGFR turnover by shifting the balance of neddylation and ubiquitination modifications on the receptor. A prediction of this model is that EGFR would fail to re-localize in response to an acute ligand stimulus in the absence of the desmosome-CSN complex. To address this, we performed immunofluorescence on NHEKs treated with 50 ng/ml of EGF ligand for 5 min to induce internalization of EGFR in the absence of Dsg1, Dp, or Cops3. As shown in *Figure 5a* (and *Figure 4—figure supplement 1*), EGFR border intensity in cells silenced for *COPS3*, *DSG1*, and *DP* remained elevated compared with control treated cultures after EGFR internalization was induced by treatment with EGF ligand. To more directly determine how neddylation regulates EGFR dynamics, we treated *NEDD8* (siNEDD8) or *COPS3* (siCOPS3) silenced NHEKs with EGF ligand to induce EGFR internalization and measured EGFR border intensity at 15 min. EGFR intensity was dramatically reduced at cell-cell interfaces in both the siCONT and siNEDD8 cultures, whereas EGFR was stable at borders in siCOPS3 cells (*Figure 5b* and *Figure 4—figure supplement 2*). These data are consistent with the idea that the desmosome-CSN complex destabilizes EGFR by de-neddylating the receptor.

To address the long-term impact of *NEDD8* knockdown on EGFR stability we treated NHEKs with cycloheximide (0.02 mg/mL) to inhibit protein translation and assessed EGFR receptor levels over time. Our results indicate that upon the loss of Nedd8, the stability of EGFR significantly decreases over a 24 hr time period (*Figure 5c*). These data are consistent with the idea that the neddylated state of EGFR promotes its stabilization. Additionally, we investigated the effect of *NEDD8* silencing on EGFR signaling. While the simple prediction of our model would be a decrease in EGFR activity,

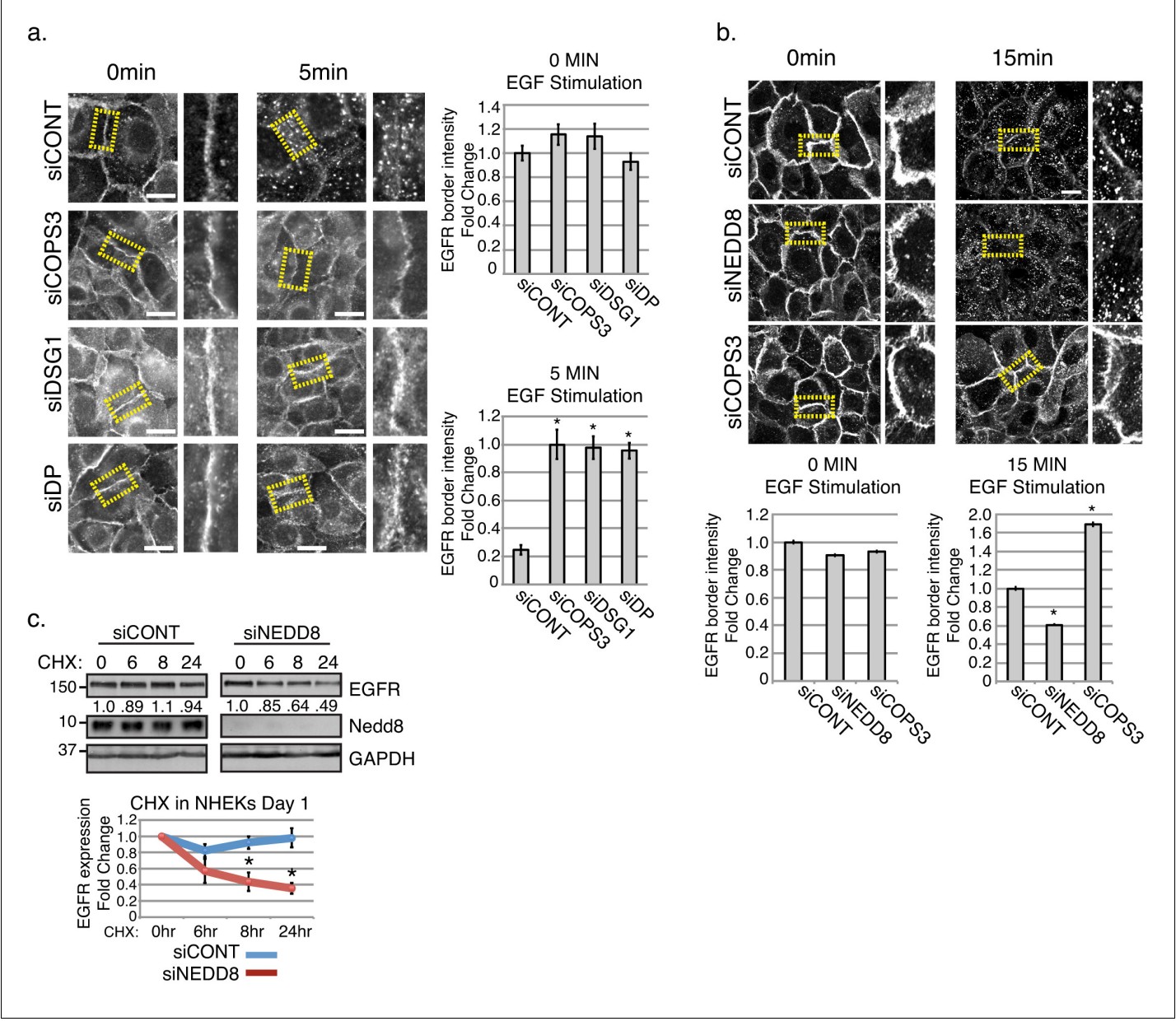

**Figure 5.** Stability of EGFR and the receptor's internalization are affected upon the loss of desmosomal components, Cops3 or Nedd8. (a) NHEKs were treated in serum free media (1.2 mM Ca++), then treated with EGF (50 ng/ml) at indicated time points to assess the amount of internalized EGFR (scale bars = 20 μm). Quantification of EGFR fluorescence border intensity from 3 independent replicates is displayed to the right of the images. *p<0.01 (Student's t test); mean ±SEM. (b) NHEKs were treated in serum free media (1.2 mM Ca++), then treated with EGF (100 ng/ml) at indicated time points to assess the amount of EGFR remaining at cell borders upon the loss of Cops3 and Nedd8 (scale bars = 20 μm). Quantification of EGFR fluorescence border intensity from 3 independent replicates is displayed below the images. *p<0.05 (Student's t test); mean ±SEM. (c) NHEKs were differentiated in high calcium for 1 day, and then treated with cycloheximide (CHX 0.02 mg/ml) for indicated times. Quantification of total EGFR is from 3 independent replicates and is normalized to GAPDH. *p<0.02 (Student's t test); mean ±SEM. All experiments are representative of 3 or more independent repeats.
DOI: https://doi.org/10.7554/eLife.22599.021

The following source data and figure supplement are available for figure 5:

**Source data 1.** EGFR localization upon stimulation in cells treated with siRNA targeting DP, DSG1, COPS3, or scramble (siCONT).
DOI: https://doi.org/10.7554/eLife.22599.023

**Source data 2.** EGFR localization upon stimulation in cells treated with siRNA targeting NEDD8, COPS3, or scramble (siCONT).
DOI: https://doi.org/10.7554/eLife.22599.024

**Source data 3.** Cycloheximide (CHX) treatment of NHEKS upon the loss of Nedd8.
DOI: https://doi.org/10.7554/eLife.22599.025

*Figure 5 continued on next page*

*Figure 5 continued*

**Figure supplement 1.** Western blot analysis of NHEKs differentiated for 1 day in 1.2 mM Ca++ with Human Keratinocyte Growth Factor (HKGF) supplement to visualize EGFR signaling at different phosphorylation sites upon the loss of Nedd8.

DOI: https://doi.org/10.7554/eLife.22599.022

we did not detect reproducible reductions in pEGFR or the activity of downstream effectors at a population level in differentiating cultures. It seems plausible that feedback mechanisms are stimulated by a complete loss of Nedd8 within these heterogeneous differentiating cultures, which could mask changes occurring at a local level (*Figure 5—figure supplement 1*).

## EGFR neddylation is elevated in squamous cell cancers and its loss accelerates keratinocyte differentiation

To address whether regulation of EGFR's neddylation is affected under conditions where EGFR expression and activity is expected to be elevated, we created 3D organotypic raft cultures using SCC9 and 1483 squamous cell cancer lines (*Figure 6a and b*). In both cases, EGFR was distributed more broadly throughout the culture and extended further into suprabasal layers than in control cultures or human epidermis. Additionally, 1483 and SCC9 3D cultures, when compared to controls, exhibited elevated levels of EGFR and Cops3 (*Figure 6a*). While we are unable to detect Dsg1 protein levels in monolayer SCC9 cultures, we have found Dsg1 to be expressed in 3D organotypic cultures that have been harvested 10 days after lifting to an air-liquid interface (*Figure 6a*). However, the level of Dsg1, and to a lesser extent Dp, is reduced in the SCC9 and 1483 cancer 3D rafts when compared to control, primary NHEK 3D rafts (*Figure 6a*). PLA analysis of organotypic cancer rafts revealed an increase in EGFR-Nedd8, which is consistent with the known elevation of EGFR activity in HNSCC (*Figure 6b* top and middle panels). Further, we observed a reciprocal staining pattern of EGFR and Dsg1 (*Figure 6b* bottom panel), consistent with the idea that loss of Dsg1 in progressing tumors is associated with elevated EGFR expression (*Wong et al., 2008*). While the HNSCC cultures displayed an increase in Cops3 protein expression, this potentially compensatory effect does not suffice to override the desmosome defect in a 3D context. To show that proper CSN functioning is required for EGFR-neddylation maintenance in a 3D model, we carried out PLA analysis of rafts silenced for *COPS3* and harvested at Day 3 (*Figure 2c*), and found that upon the loss of Cops3 there is an increase in Nedd8 associated with EGFR (*Figure 6d*).

## Discussion

Here we demonstrate a novel form of EGFR regulation and turn over by a newly described desmosome-CSN cell junctional complex. We propose a model by which the desmosome scaffolds the CSN in proximity to EGFR, to control the balance of neddylation-ubiquitination modifications, thus affecting receptor dynamics. Our data is consistent with a model whereby the CSN de-neddylation function contributes to EGFR turnover to promote differentiation in a regenerating tissue, in this case, the epidermis.

Previous work has shown that the loss of one CSN component affects the formation and function of the entire CSN (*Denti et al., 2006*; *Wei and Deng, 1999*). Thus, silencing of the *COPS3* subunit, shown here to interact with the desmosome, is predicted to compromise the entire CSN. Similarly, silencing either *DP* or *DSG1* disrupts the association of Cops3 with the other junction component, consistent with the idea that entire desmosome is involved in scaffolding the CSN. This idea is further supported by a mass spectrometry study that identified multiple desmosomal proteins (Plakophilin2 (Pkp2), Pg, Dsg1, and Dp), and not adherens junction proteins, as putative interactors with the first subunit (Cops1) of the COP9 signalosome (*Fang et al., 2012*). The fact that desmosomal control over the neddylation-ubiquitination balance occurs in SCC9s and keratinocytes in the absence of Dsg1 supports the idea that desmosomes of varying composition broadly regulate the CSN.

Our data provide further support for a functional role of Nedd8 modifications on less commonly reported non-cullin substrates, in this case EGFR. Further, our findings are consistent with the idea that neddylation of EGFR stabilizes the receptor (*Figure 5c*) and prevents EGFR from ligand-induced internalization (*Figure 5a and b*). While one study reported a correlation between EGFR neddylation

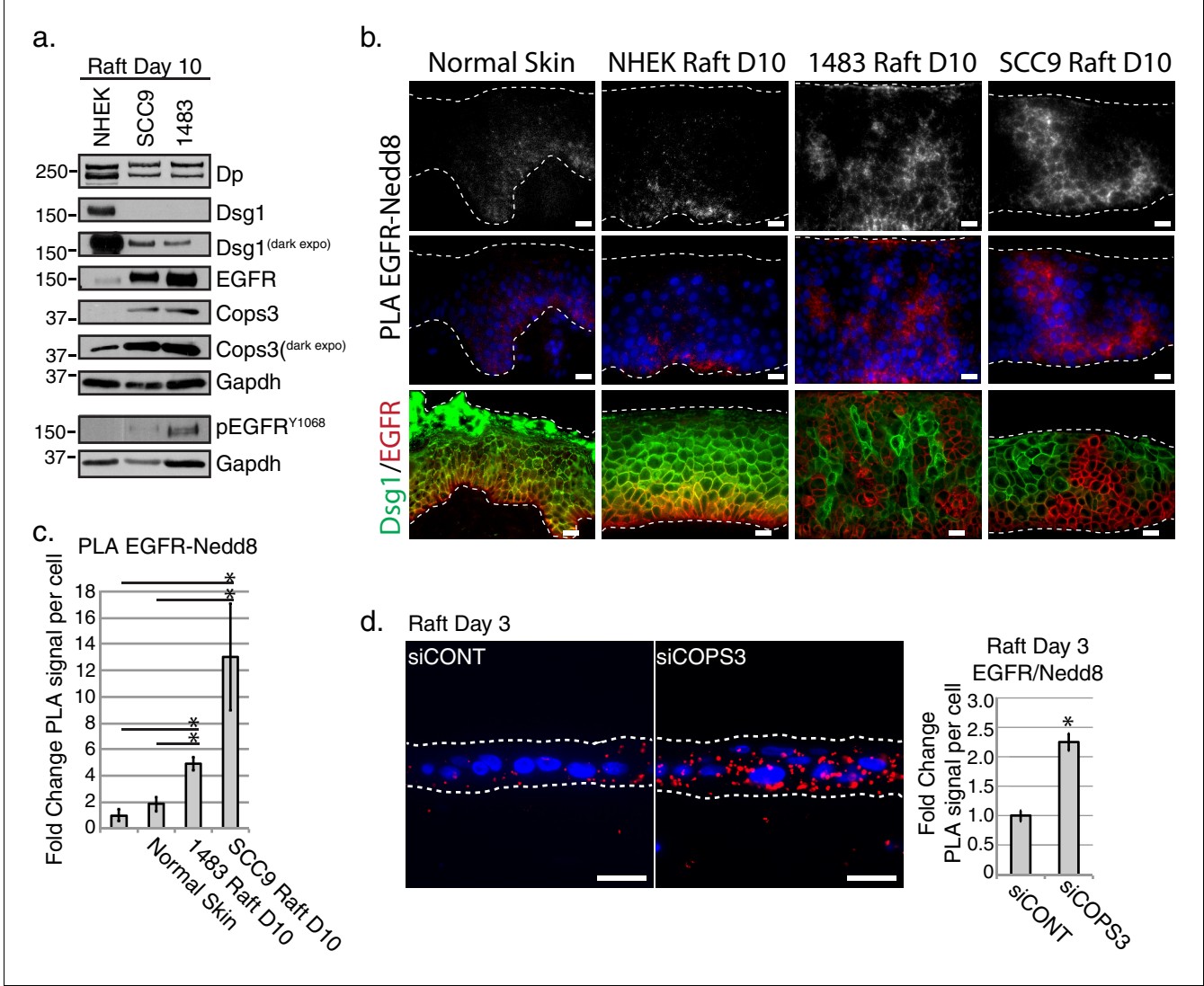

**Figure 6.** EGFR neddylation is increased in 3D Head and Neck Squamous Cell Carcinoma (HNSCC) and Cops3 knockdown organotypic models. (**a**) Biochemical analyses of HNSCC 3D organotypic raft cultures generated using SCC9 and 1483 cell lines displayed a decrease in Dsg1 and Dp, and an increase in EGFR and Cops3. (**b**) PLA analysis of EGFR[RABBIT] and Nedd8[MOUSE] in normal skin, and 3D organotypic raft cultures generated using the NHEKs, SCC9s, and 1483 HNSCC cell lines (scale bar = 20 µm). Top panel displays PLA signal alone with an apparent cell border signal, and middle panel represents the PLA signal with Dapi overlay. Bottom panel displays fluorescent staining of Dsg[GREEN] and EGFR[RED] in normal skin, and NHEK, SCC9 or HNSCC rafts. (**c**) Quantification of PLA signals in 3D cancer models. *p<0.05 (Student's t test); mean ±SEM. (**d**) PLA analysis of 3D organotypic raft cultures generated with NHEKs treated with siRNA scramble sequence (siCONT) or siRNA directed towards Cops3 (siCOPS3), indicating an increase in EGFR-Nedd8 PLA signal (scale bar = 20 µm). Quantification of PLA signals displayed to the right of the images. *p<0.05 (Student's t test); mean ±SEM. All experiments are representative of 3 or more independent replicates.

DOI: https://doi.org/10.7554/eLife.22599.026

The following source data is available for figure 6:

**Source data 1.** PLA analysis Nedd8/EGFR in 3D cultures.
DOI: https://doi.org/10.7554/eLife.22599.027

**Source data 2.** PLA analysis of Nedd8/EGFR 3D organotypic deficient in Cops3.
DOI: https://doi.org/10.7554/eLife.22599.028

and degradation in Chinese hamster ovary (CHO) cells, the majority of studies have linked neddylation with an increase in protein stability (e.g. MDM2 (*Watson et al., 2010*), L11 (*Sundqvist et al., 2009*), HIF1α (*Ryu et al., 2011*), and TβRII (*Zuo et al., 2013*). It is unknown whether specific neddylated residues on EGFR are important for driving EGFR towards certain outcomes or whether the

number of Nedd8 modifications on EGFR is important. Poly-Nedd8 chains have been demonstrated, but it is still unclear whether these chains have a function *in vivo* (*Jones et al., 2008*; *Xirodimas et al., 2008*). Nevertheless, our data showing a reciprocal relationship between neddylated and ubiquitinated EGFR suggest that Nedd8 and Ubiquitin compete for lysine residues on EGFR to regulate receptor dynamics. Furthermore, our data demonstrate that EGFR ubiquitination can occur independent of neddylation (*Figure 4e* and *Figure 4—figure supplement 2*), suggesting a new level of regulation that could have broad-reaching implications for receptor-mediated signaling in multiple contexts.

While we show changes in the neddylation state of EGFR, we have not ruled out the possibility that the desmosome-CSN super complex also regulates other proteins at the membrane, such as better-known substrates for the CSN, the cullin proteins, which could promote local Ubiquitin activity at the membrane. It is possible that a uniquely targeted protein degradation program could work in concert with transcriptional epidermal differentiation programs to remove proteins that inhibit, or are no longer required for, epidermal differentiation. Such a process could contribute to the resculpting and polarization of cell membrane proteins known to occur during differentiation (*Niessen et al., 2012*; *Wollner et al., 1992*).

In addition to a role for desmosomal scaffolding of the CSN in regulating normal epidermal differentiation and homeostasis, our data from a 3D organotypic cancer model suggests that EGFR neddylation may contribute to elevated EGFR/MAPK signaling in human diseases including inherited keratodermas (*Harmon et al., 2013*) and cancers such as head and neck squamous cell carcinoma (HNSCC) (*Figure 6a–c*). EGFR has been shown to be a strong prognostic indicator in epithelial cancers, and increased expression has been associated with reduced survival rates in 70% of studies (*Nicholson et al., 2001*). This suggests the possibility that elevation of EGFR in different types of cancer may be partly due to the stabilization of the receptor through increased neddylation. Additionally, Cops3 has been implicated in keratinocyte interleukin signaling (*Banda et al., 2005*), which is interesting in light of the recent observation that loss of Dsg1 leads to an increase in inflammatory mediators in SAM syndrome, featuring severe dermatitis, multiple allergies, and metabolic wasting (*Samuelov et al., 2013*). A desmosome-CSN super complex could thus serve as a general mechanism by which desmosomes regulate multiple downstream functions that guide normal skin homeostasis and paracrine signaling pathways in which keratinocytes participate. Further, components of this super complex could provide new targets for treating diseases associated with inappropriate epidermal differentiation, inflammation, and cancer.

## Materials and methods

### Cell culture and drugs

Primary normal human epidermal keratinocytes (NHEKs) were cultured in M154 media (Invitrogen) adjusted to 0.07 mM calcium (termed as low calcium) and supplemented with human keratinocyte growth supplement (HKGS) and gentamicin/amphotericin B. To induce differentiation, NHEKs were switched to media with 1.2 mM calcium (termed as high calcium). High calcium media did not contain HKGS except where noted in text. Human-derived oral squamous cell carcinoma SCC9 cells (RRID: CVCL_1685) and 1483 (RRID: CVCL_6980) were maintained in DMEM/F-12 medium (Mediatech) supplemented with 10% FBS and 1% penicillin/streptomycin. EGFR inhibitor, AG1478 (Selleck Chemicals) was used at a final concentration of 2 μM. Cycloheximide (CHX) was used at a final concentration of 0.02 mg/ml.

### Cell authentication

Primary normal human epidermal keratinocyte isolates (NHEKs) are obtained through the Northwestern University Skin Disease and Research Core, where mycoplasma, HIV-1, hepatitis B and C testing is routinely performed. Primary keratinocyte purity is assessed by immunostaining for epidermal keratinocyte specific markers, such as keratins K1/K10 and K5/K14. Cell lines are routinely confirmed to be mycoplasma negative using the Lonza MycoAlert mycoplasma detection Kit and/or by real-time PCR (IDEXX BioResearch (Columbia, MO). The SCC9 and 1483 lines were analyzed by short tandem repeat (STR) profiling to detect both contamination and misidentification, including intra- and inter-species contamination by IDEXX BioResearch (Columbia, MO). The SCC9 cell line (RRID: CVCL_

1685, gift from J. Rheinwald, Harvard Medical School, Boston, MA) scored above 80% indicating the sample is consistent with the cell line of origin. The 1483 cell line (RRID: CVCL_6980, a human HNSCC line derived from the oropharynx obtained from Jennifer Grandis) is not available from ATCC. STR analysis ruled out inter-species contamination. While the line scored less than an 80% match compared with the IDEXX standard for STR analysis, this standard was shown to be contaminated with the UM-SCC-1 line (*Zhao et al., 2011*). Thus, key features of this line critical for the present work were validated independently through analysis of EGFR expression, response to inhibitors, and assessment of HNSCC keratins and junctional proteins, including Dsg1 whose expression is lost in many of the available HNSCC lines (*Thomas et al., 2008*; Desai BD, Todorovic V, and Green KJ, unpublished).

## Antibodies

The following primary antibodies were used in this study: Cell Signaling: rabbit anti-HA (RRID:AB_1549585), rabbit anti-EGFR D38B1 (RRID:AB_2246311), rabbit anti-pEGFR$^{Y1045}$ (RRID:AB_331710), anti-pEGFR$^{Y1068}$ (RRID:AB_2096270), anti-pEGFR$^{Y1086}$ (RRID:AB_823485), rabbit anti-pErk42/44 (RRID:AB_2315112), rabbit anti-pMek (RRID:AB_2138017), rabbit anti-Mek (RRID:AB_823567) rabbit anti-ErbB2 (RRID:AB_10692490), rabbit anti-pErbB2 (RRID:AB_490899). Abcam: rabbit anti-Cops3 (RRID:AB_1603748), rabbit anti-Nedd8 (RRID:AB_1267251). Sigma-Aldrich: M2 mouse anti-FLAG (RRID:AB_259529), rabbit anti-FLAG (RRID:AB_439687), rabbit anti-GAPDH (RRID: AB_796208), mouse anti-HA (RRID:AB_262051), C2206 rabbit anti-Beta-catenin (RRID:AB_476831), mouse anti-Nedd8 (RRID:AB_260757). Progen: U100 mouse anti-Dsc1 (Cat#65192), p124 mouse anti-Dsg1(Cat # 651111). Invitrogen: 27B2 mouse anti-Dsg1 (RRID:AB_2533088). NeoMarkers: AB-12 mouse anti-EGFR (Cat # MS-400-P1). GE Healthcare Biosciences: goat anti-GST (RRID:AB_771432). Enzo Life Sciences: rabbit anti-Nedd8 (RRID:AB_2051982). Millipore: FK2 mouse anti-Ub (RRID:AB_612093). Aves Laboratories: 1407 chicken anti-Pg. R&D Systems: goat anti-Dsg1 (RRID:AB_2277393). NW161 and NW6 rabbit anti-desmoplakin (*Bornslaeger et al., 1996*). Promega: Anti-Erk 1/2 (Cat #V114a). Santa Cruz: mouse anti-Cops3 (RRID:AB_2081616). Gift antibodies: 1G4 mouse anti-Dp (gift from J. Wahl III, University of Nebraska, Omaha, NE, USA), rabbit anti-K1, rabbit anti-K10, rabbit-anti-Lor (gifts from J. Segre National Human Genome Research Institute, Bethesda, Maryland, USA), 1G5 mouse anti-Alpha-catenin (gift from Margaret Wheelock, University of Nebraska, Omaha, NE, USA, in memoriam), HecD-1 mouse anti-Ecad (gift from M. Takeichi and O. Abe, Riken Center for Developmental Biology, Kobe, Japan). Western blot analysis included use of peroxidase-conjugated anti-mouse, -rabbit, and -chicken secondary antibodies purchased from SeraCare Life Sciences, Baltimore, MD (formerly Kirkegaard and Perry Laboratories). AlexaFluor 488/568/647-conjugated goat anti- mouse, -rabbit, and –chicken secondary antibodies (Invitrogen) were used in immunofluorescence studies.

## Generation and delivery of DNA constructs

The Dsg1 cytyoplasmic tail was cloned into a bait construct (pSos-Dsg1) (constructs provided by Invitrogen). The Dsg1-FLAG and Dsg1-GFP construct was generated as previously described (*Getsios et al., 2009*; *Harmon et al., 2013*). The HA-Nedd8 construct was obtained from Addgene (plasmid #18711) and transiently transfected using Lipofectamine 2000 (Invitrogen) 2–15 µg/µl DNA.

## Retroviral infections

The phoenix packaging cell line (provided by G. Nolan, Stanford University, Stanford California, USA), maintained in DMEM (Mediatech) supplemented with 10% FBS and 1% penicillin/streptomycin, was transfected with LZRS constructs (0.5–2 µg/ml). The day after transfection, cells were re-seeded into selection media with 1 µg/ml puromycin. Once cells reached 80% confluency, they were switched to 32°C for 24 hr to collect viral supernatant. In some cases viral supernatant was used immediately to infect primary keratinocytes, and in other cases viral supernatant was concentrated using Amicon Ultra-15 Centrifugal Filter Units (Millipore). NHEKs were infected with virus and 8 µg/ml polybrene for 90–180 min at 32°C, after which cells were washed and returned to growth at 37°C in fresh media.

## siRNA-mediated knockdown

NHEKs and SCC9s were electroporated with siRNA oligonucleotides at a final concentration of 10 nM via AMAXA nucleoporation (Lonza) using the Ingenio Electroporation Solution (Mirus) or solution V (Lonza) and program X-001. siRNA directed towards *DSG1* (5' CCA UUA GAG AGU GGC AAU AGG AUG A), *DP* (Invitrogen Pool: oligonucleotides: 5'-GAA GAG AGG UGC AGG CGU A, 5' GAC CGU CAC UGA GCU AGU A, and 5' AAA CAG AAC GCU CCC GAU A), and *COPS3* (Invitrogen Stealth siRNA_siCOPS3-465: 5' AUC AAU GUU AUG AAG CAU GGC UGG G or Integrated DNA Technologies Pool_IDT Pool: 5'-GGA UAU CUG UAA AGA GAA UGG AGC C, 5'-GGA UGUACA AGA ACA CUC CUU GGG C, 5'-UCC AUC CUG AGC UAA ACA AGA GAA A, and Dharmacon (D-011494–03)_siCOPS3 Dharma3: 5'- UCC GAA ACC UGG UGA AUA A), *NEDD8* (Dharmacon siGE-NOME SMARTpool (M-020081–01): 5'-GAA AGG AGA UUG AGA UUG A, 5'-AGA UUG AGA UUG ACA UUG A, 5'-CAG ACA AGG UGG AGC GAA U, 5'-GGA GAU UGA GAU UGA CAU U), and scramble/non-targeting siRNA (Dharmacon D-001206-14-20).

## Western blot analysis

For analysis of protein expression levels, cells were washed in phosphate buffered saline (PBS) and lysed in urea sample buffer (8 M deionized urea, 1% sodium dodecyl sulfate, 10% Glycerol, 60 mM Tris pH 6.8, and 5% β-mercaptoethanol). After equalizing total protein concentrations, samples were run on 7.5–15% SDS-PAGE gels and transferred to polyvinylidene fluoride (PVDF) or nitrocellulose membranes to be probed with primary and secondary antibodies against proteins of interest.

## Immunofluorescence and immunohistochemistry

For analysis of protein localization in cultured cells, NHEK coverslips were fixed and permeabilized by submerging in anhydrous methanol for 2–3 min at −20°C when visualizing Dp. All other antibodies were applied to NHEKs fixed and permeablized in paraformaldehyde followed with a 10 min 0.2% Triton X100 wash at 4°C. Cells were incubated in solutions with primary antibodies at 4°C overnight, and secondary antibodies at 37°C for 30–60 min, with multiple PBS washes after each step. For analysis of protein localization in tissue sections, paraffin sections of organotypic raft cultures were rehydrated and then heated to 95°C in 0.01 M citrate buffer for antigen retrieval. Sections were incubated in solution with primary antibody at 4°C overnight and in solution with secondary antibody at 37°C for 60 min. Coverslips and tissue sections were mounted on polyvinyl alcohol (Sigma-Aldrich).

## Microscope image acquisition

Cells and tissues were visualized using a Leica microscope (model DMR, Melville NY) fitted with 40X (PL Fluotar, NA 1.0) and 63X (PL APO, NA 1.32) objectives. Images on the DMR microscope were acquired using an Orca 100 CCD camera (model C4742-95; Hamamatsu, Bridgewater, NJ) and analyzed using ImageJ software (NIH version 2.0.0) or Metamorph. For N-SIM analysis, the samples were illuminated with spatially high-frequency patterned excitation light (100X objective lens, NA 1.49; TiE N-SIM microscope [Nikon] and iXON X3 897 camera [Andor Technology]). Images were reconstructed and analytically processed to reconstruct subresolution structure of the samples using Elements version 4 software (Nikon). For siNEDD8 and siCOPS3 EGFR internalization, the Axioimager and the Apotome2 was used with the 40x objective and a MIP of the Z sections was used.

## Organotypic raft cultures

3D raft cultures were grown following protocols previously described (*Simpson et al., 2010*). Collagen plugs were formed using rat-tail collagen type I (BD) and seeded with J2 fibroblasts (NIH 3T3). J2 fibroblasts were cultured in DMEM (Mediatech) supplemented with 10% FBS and 1% penicillin/streptomycin (Mediatech). E-media was made as a 3:1 mix of DMEM/F-12 and DMEM media with 5% FBS, 10 µg/ml gentamicin (Life Technologies), 0.4 µg/ml hydrocortisone (Sigma), 10 ng/ml cholera toxin (Sigma), and 0.25 µg/ml amphotericinB (Life Technologies), mixed with a cocktail of 180 µM adenine (Sigma), 5 µg/ml human recombinant insulin (Sigma), 5 µg/ml human apo-transferrin (Sigma), and 5 µg/ml triiodothyronine (Sigma). After seeding with NHEKs, cancer cell lines (SCC9, 1483), or NHEKs that had been electroporated with control or Cops3 siRNA (Amaxa), rafts were grown in E-media with 5 ng/ml epidermal growth factor (EGF) (EMD Millipore). Two days later, rafts

were lifted to an air liquid interface and grown in E-media without EGF to induce epidermal differentiation, and harvested on the indicated number of days.

## Yeast-two-hybrid screening (Y2H)

Y2H screening was performed following the CytoTrap vector kit (Stratagene) as described (*Harmon et al., 2013*). The CytoTrap screen (Stratagene) utilizes the temperature-sensitive cdc25H strain of *S. cerevisiae*, which is deficient in Ras signaling and unable to grow at 37°C. While yeast grow at the permissive temperature of 25°C, only yeast that are co-transformed with a bait (pSos) construct that binds a membrane-associated myristoylated target (pMyr) protein activate Ras signaling to allow for growth at 37°C. pMyr constructs are expressed under galactose promoters; thus, growth at 37°C on galactose plates without growth at 37°C on glucose plates indicates target-bait interaction (growth on glucose plates at 37°C indicates temperature sensitive revertants). A library of HeLa cell cDNAs expressed in the pMyr vector was co-transformed with pSos-DP-NTP or pSos-Dsg1 into *cdc25*H yeast. After obtaining putative positives by identifying clones that were capable of growing on galactose plates at 37°C for interactions with Dsg1, pMyr constructs were purified and individually co-transformed with pSos-Dsg1 to confirm specific interactions. pMyr-Pg was used as a positive binding control for Dsg1, and pMyr and pMyr-SB (Stratagene) were respectively used as negative and positive controls for the screen.

## Glutathione-S-Transferase

Dsg1-GST and GST constructs were bacterially expressed by adding isopropyl β-D-1-thiogalactopyranoside (IPTG) to BL21A1 bacteria (Invitrogen). Bacteria were lysed and proteins were purified using glutathione agarose (GE Healthcare). Beads were then incubated with NHEK cell lysates at 4°C and washed with buffer (500 mM NaCl, 50 mM Tris pH 7.6, 10 mM MgCl$_2$, 1% Tx-100, 0.1% SDS, 0.5% deoxycholate) and eluted in urea sample buffer or Laemmli buffer with 5% β-mercaptoethanol.

## Co-immunoprecipitation studies

S-tag immunoprecipitations were performed as previously described (*Albrecht et al., 2015*). Cells used for co-immunoprecipitation studies were rinsed twice in PBS on ice and lysed in ice-cold 1.0 ml RIPA buffer (500 mM NaCl, 50 mM Tris pH 7.6, 10 mM MgCl$_2$, 1% Tx-100, 0.1% SDS, 0.5% deoxycholate) with complete protease inhibitor cocktail (Roche). Cells were incubated with 0.5–1.5 μg of antibody against the protein of interest overnight at 4°C. The immunoprecipitate was then conjugated to Protein A/G beads (Santa Cruz sc-2003) and eluted in urea sample buffer or Laemmli/Sample buffer (200 mM Tris pH6.8, 20% glycerol, 4% SDS, 0.3% bromophenol blue) with 5% β-mercaptoethanol.

## EGFR immunoprecipitations

SCC9s and NHEKs used in EGFR immunoprecipitations were rinsed twice with PBS on ice and lysed in ice-cold 1.0 mL RIPA buffer (50 mM Tris H-Cl pH 7.5, 150 mM NaCl, 1 mM EDTA, 1% Triton X-100, 1% sodium deoxycholate, 0.1% SDS) with phosphatase inhibitors (1:100 Millipore Phosphatase Inhibitor Cocktail Set II (524625) and 1:50 Millipore Phosphatase Inhibitor Cocktail Set IV (524628)) and Complete EDTA-free protease inhibitor cocktail (Roche 05056489001). Cells were incubated with EGFR antibody (1 μg/mL) from Cell Signaling (D38B1) for an overnight rotation at 4°C. 15 μL of Protein A/G PLUS-agarose beads (Santa Cruz sc-2003) were added the next day for 45 min at 4°C. Beads were then washed in the RIPA buffer used for lysis, and proteins were eluted off beads using 30 μl of Laemmli lysis buffer. The entire lysate was loaded onto a SDS-PAGE gel for western blot analysis. To assay whether ubiquitination required neddylation for EGF-bound receptors, we performed EGFR IPs in SCC9s silenced for NEDD8 and serum starved, and treated these cells with EGF ligand (50 ng/mL) for 5 min.

## Proximity ligation assay (PLA)

Reagents used to conduct PLA were purchased from DUOLink Biosciences and used as described (*Harmon et al., 2013*). Cells were rinsed and fixed as described above. After incubation with primary antibody overnight at 4°C, samples were incubated with PLA secondary antibodies conjugated to DNA oligonucleotides for 60 min at 37°C. Samples were then subjected to a 30 min incubation at

37°C for ligation of nucleotides, followed by a 100 min incubation at 37°C for rolling circle polymerization, resulting in the production of fluorescent dots (shown in yellow or red) if the antigens targeted by secondary antibodies were in close proximity (40–100 nm). ImageJ software or Metamorph was used to quantify the number of PLA and DAPI signals per image field.

### Sample size and statistical analysis

All experiments were performed independently $\geq$3 times (i.e. biological replicates performed on different days, not technical replicates performed in parallel at the same time). For each independent/ biological replicate, multiple experimental/control arms were processed and analyzed in parallel. Representative experiments are displayed throughout the figures and where indicated in the figure legends, quantification of experiments is reported as mean ±standard error mean (SEM). For biochemical analyses, cell lysates were derived from a population of cells grown in 10 cm dishes (approximately 4 million cells) or 3 cm dishes (approximately 1 million cells). For immunofluorescence or PLA analyses, cells were plated onto a coverslip in a 3 cm dish and quantified as follows: PLA particles were counted for 5–10 randomly selected fields, each containing 20–100 cells (depending on whether image was obtained using a 40X or 63X objective and/or whether the image was from cultured cells or 3D organotypic rafts). Parallel analyses of protein specific antibodies with IgG (mouse or rabbit) were performed to control for false positives. Fluorescence pixel intensity at randomly selected cell borders was determined by multiplying the mean pixel intensity by the area of the defined border divided by the border length. Background intensity was randomly selected from an area on the image and subtracted from the border intensity. Two group comparisons were performed using two-tailed, two-sample equal variance Student's t test using Excel (Microsoft, Redmond, WA, USA). p-values<0.05 were considered statistically significant. Densitometric analyses were performed on scanned films of immunoblots with the lightest possible exposure, and numbers derived from the densitometric analyses were displayed below immunoblots as indicated in the figures. All densitometric quantifications were normalized to a protein standard, as indicated in figures. All immunofluorescence and densitometric calculations were performed using FIJI (FIJI Is Just ImageJ) (*Schindelin et al., 2012*), ImageJ (*Schneider et al., 2012*) or Metamorph.

## Acknowledgements

The authors would like to thank all the members of the Green Laboratory, Cory Simpson, M.D., Ph. D., Robert Lavker, Ph.D., and Spiro Getsios, Ph.D. for insightful discussions during development of this project and comments on the manuscript. Gift antibodies were provided by Julie Segre (K1, K10, Loricrin). Keratinocytes were obtained from the Keratinocyte Core of the Northwestern University Skin Disease Research Center, and histological analysis (H and E staining) was conducted by the Pathology Core of the Northwestern Skin Disease Research Center Chicago, IL, USA, with support from the National Institute of Arthritis And Musculoskeletal And Skin Diseases of the National Institute of Health (NIAMS/NIH) (P30AR057216). Nikon SIM imaging was performed at the Northwestern University Cell Imaging Facility supported by the National Cancer Institute of the National Institute of Health (NCI/NIH) grant CCSG P30 CA060553 (Robert H Lurie Comprehensive Cancer Center). This work was supported by NIAMS/NIH (R01 AR041836 and R37 AR043380, with partial support from R01 CA122151). NA Najor was supported by a Kirschstein postdoctoral fellowship (F32AR066465). LV Albrecht was supported by a Kirschstein predoctoral fellowship (T32GM008061), by the Malkin Scholar Program from the Robert H Lurie Comprehensive Cancer Center of Northwestern University, and an American Heart Association predoctoral fellowship. RM Harmon was supported by a Kirschstein predoctoral fellowship (T32GM008061) and an American Heart Association predoctoral fellowship. Any opinions, findings, and conclusions or recommendations expressed in this material are those of the author(s) and do not necessarily reflect the views of the Northwestern University Skin Disease Research Center, NIAMS/NIH, or NCI/NIH.

## Additional information

### Funding

| Funder | Grant reference number | Author |
|---|---|---|
| National Institute of Arthritis and Musculoskeletal and Skin Diseases | Postdoctoral Fellowship (F32AR066465) | Nicole Ann Najor |
| National Institute of General Medical Sciences | Graduate Student Training Grant (T32GM008061) | Robert Harmon Lauren Veronica Albrecht |
| American Heart Association | Predoctoral Fellowship | Lauren Veronica Albrecht Robert Harmon |
| National Institute of Arthritis and Musculoskeletal and Skin Diseases | Research Program Grant (R01 AR041836) | Kathleen Janee Green |
| National Institute of Arthritis and Musculoskeletal and Skin Diseases | Research Program Grant (R37 AR043380) | Kathleen Janee Green |
| National Cancer Institute | Research Program Grant (R01 CA122151) | Kathleen Janee Green |

The funders had no role in study design, data collection and interpretation, or the decision to submit the work for publication.

### Author contributions

Nicole Ann Najor, Conceptualization, Formal analysis, Funding acquisition, Validation, Investigation, Visualization, Methodology, Writing—original draft, Project administration, Writing—review and editing; Gillian Nicole Fitz, Formal analysis, Validation, Investigation, Methodology, Writing—review and editing; Jennifer Leigh Koetsier, Validation, Investigation, Writing—review and editing; Lisa Marie Godsel, Formal analysis, Validation, Investigation, Writing—review and editing; Lauren Veronica Albrecht, Investigation, Methodology, Writing—review and editing; Robert Harmon, Investigation, Methodology, Performed the initial yeast-2-hybrid screen, which revealed the Cops3 and Dsg1 interaction; Kathleen Janee Green, Conceptualization, Resources, Supervision, Funding acquisition, Methodology, Project administration, Writing—review and editing

### Author ORCIDs

Nicole Ann Najor http://orcid.org/0000-0002-1510-9979
Gillian Nicole Fitz http://orcid.org/0000-0002-5995-6918
Kathleen Janee Green http://orcid.org/0000-0001-7332-5867

### Decision letter and Author response

Decision letter https://doi.org/10.7554/eLife.22599.033
Author response https://doi.org/10.7554/eLife.22599.034

## Additional files

### Supplementary files

• Transparent reporting form
DOI: https://doi.org/10.7554/eLife.22599.029

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
