## [Decision Letter]

Thank you for submitting your article "EGFR neddylation is regulated by a novel desmosomal–COP9 signalosome complex" for consideration by *eLife*. Your article has been favorably evaluated by Fiona Watts (Senior editor) and three reviewers, one of whom, Reinhard Fässler (Reviewer 1), is a member of our Board of Reviewing Editors. The following individuals involved in review of your submission have agreed to reveal their identity: George Garinis (Reviewer #2); Markus Schober (Reviewer #3).

The reviewers have discussed the reviews with one another and the Reviewing Editor has drafted this decision to help you prepare a revised submission.

Summary:

Desmoglein 1 (Dsg1) was shown to inhibit Epidermal growth factor receptor (EGFR) activity resulting in cell cycle exit and epidermal differentiation. How Dsg1 controls EGFR activity is unknown. The authors report now that the desmosomal components Dsg1 and Dp in keratinocytes recruit the COP9 signalosome (CSN) to de–neddylate EGFR resulting in increased ubiquitination and turnover of EGFR, and keratinocyte differentiation. These findings are highly significant and shed new light on the regulation of EGFR during epidermal differentiation. However, a potential conflict with previous reports on the role of EGFR neddylation for ubiquitylation should be resolved and important controls included into a revised manuscript. The essential revisions and the minor issues follow.

Essential revisions:

1) In contrast to the authors' finding that EGFR neddylation allows signaling and EGFR ubiquitylation leads to degradation, the Yarden lab (Oved et al., 2006) reported that EGFR neddylation is required for the subsequent ubiquitylation and degradation of activated (EGF–bound) EGFR. The potential contradiction between the separation versus dependence of these events should be solved.

2) What happens with EGFR stability and signaling when Nedd8 is depleted with specific siRNAs?

3) The siCOPS3 should be rescued with an siRNA resistant COPS3 cDNA.

4) The authors perform several PLA assays to illustrate direct protein interactions in vivo. Unfortunately, PLA assays are prone to give false positive results and the data would be more convincing if another junctions protein that does not physically interact with Dsg1/DP was used as a negative control for the experiment. Along these lines, the signal distribution in the SIM and PLA experiments appear different. It would help if the authors could provide phase contrast or junctions marker staining to better illustrate cell–cell adhesion in the PLA data.

---

## [Author Response]

Essential revisions:1) In contrast to the authors' finding that EGFR neddylation allows signaling and EGFR ubiquitylation leads to degradation, the Yarden lab (Oved et al., 2006) reported that EGFR neddylation is required for the subsequent ubiquitylation and degradation of activated (EGF–bound) EGFR. The potential contradiction between the separation versus dependence of these events should be solved.

We agree with the reviewer that the extent to which separation versus dependence of ubiquitination (ubiquitylation) and neddylation of the EGFR receptor is important in terms of further understanding EGFR dynamics. To clarify the relationship between neddylation and ubiquitination in human keratinocytes, we performed EGFR immunoprecipitations in SCC9s that have been silenced with siRNA directed towards *NEDD8*, starved, and then stimulated with EGF. As shown in Figure 4, in spite of virtually complete loss of Nedd8, EGFR remains ubiquitinated, providing direct support that EGFR neddylation is not required for its ubiquitination in keratinocytes. Additionally, PLA analysis demonstrated that while PLA signals for the EGFR–Nedd8 pairing were negligible upon loss of Nedd8, ubiquitin remained robustly associated with EGFR. This result has been included as Figure 4—figure supplement 2. We have adjusted the text to include the new data (Figure 4, Figure 4—figure supplement 2, and subsection “EGFR neddylation is elevated in desmosome and Cops3 deficient cells”).

In addition to remaining ubiquitinated, EGFR stability was decreased in Nedd8 silenced cells following a cycloheximide chase (see essential revisions #2, below). Further, as shown in new figure panel, Figure 5, upon an acute EGF stimulus, EGFR exhibited a more rapid redistribution away from cell–cell interfaces as measured by loss of fluorescence intensity in Nedd8 knockdown keratinocytes, consistent with more rapid internalization in the absence of Nedd8. Collectively, these data provide direct evidence that in human keratinocytes, Nedd8 is not required for ubiquitination and turnover of activated EGFR, and that neddylation stabilizes the receptor.

These data are a departure from those reported by (Oved et al., 2006) in which it was concluded that EGFR neddylation and ubiquitination occur on competing lysines in CHO cells to drive turnover of the receptor. As described above, our data support the idea that EGFR neddylation stabilizes the receptor. Possible reasons for differences between the two studies include a) the use of different cell types and b) differences associated with analysis of the dynamics of ectopic EGFR (Oved et al., 2006) and endogenous EGFR (current work). The research performed by Oved et al., 2006, studied turnover of ectopically expressed EGFR in CHO cells, which do not endogenously express the receptor. In our manuscript we are studying dynamics of the endogenous EGFR in human primary keratinocytes and epithelial cell lines, in which EGFR activity and stability are subject to a complex regulatory network that assists to control the balance of proliferation and differentiation. While it is likely that some aspects of EGFR regulation are shared (for instance, our data show a reciprocal relationship between neddylation and ubiquitination, thus indirectly supporting the idea that these modifications may compete for the same lysines on EGFR in keratinocytes), mechanisms regulating the dynamics of EGFR modification and turnover are likely to exhibit biologically significant differences. Collectively, our data provide direct evidence that depleting the system of endogenous Nedd8 does not prevent ubiquitination of endogenous EGFR in keratinocytes, and that loss of Nedd8 destabilizes the receptor.

2) What happens with EGFR stability and signaling when Nedd8 is depleted with specific siRNAs?

To address this comment we carried out a cycloheximide chase and quantified total EGFR over time in keratinocytes treated with siRNA directed towards *NEDD8* (siNEDD8) or a scramble oligo (siCONT). We found that upon silencing of *NEDD8* there is a significant decrease in EGFR levels over time (*p < 0.02), further supporting our hypothesis that the neddylated state of the receptor is more stable. We have adjusted Figure 5 to include the cycloheximide chase data and the quantification from three independent repeats (Figure 5). We have also adjusted the text to reflect this supporting evidence (subsection “EGFR neddylation is elevated in squamous cell cancers and its loss accelerates keratinocyte differentiation” and Discussion section).

Additionally, we investigated the signaling effects in NHEKS upon the loss of Nedd8. While the simple prediction of our model would be a decrease in EGFR activity, we did not detect reproducible reductions in EGFR phosphorylation or the activity of downstream effectors at a population level in differentiating cultures. The data is included as Figure 5—figure supplement 1. A population analysis of primary keratinocytes, silenced for *NEDD8* (using a pool of oligos as well as four individual oligos) and differentiated for two days in 1.2mM Ca^2+^ media, revealed the complexity of changes within these heterogeneous cultures (see Author response image 1). Through visualization of the differentiation marker, Dsg1, and Dapi–positive staining nuclei, we saw that overall there were more cells in the Nedd8 knock down cultures (*p<0.05) and a significant population had initiated the differentiation program (*p<0.05), which would be predicted to occur when EGFR activity is suppressed (bar graphs to right). However, due to the large number of cells in the cultures that were still undifferentiated, changes in differentiation markers on a population level assessed by western blot did not allow for appreciation that these cultures were robustly differentiating. It seems likely that feedback mechanisms stimulated in response to complete loss of Nedd8 may mask changes that occur at a local level such that on average there is no apparent change in signaling. The observed increase in differentiation is shown below for the reviewers’ information, but not included in the paper as while multiple Nedd8 oligo targets were used, the experiment was only performed once.

3) The siCOPS3 should be rescued with an siRNA resistant COPS3 cDNA.

We acknowledge the reviewers concern related to off target effects. We took a multi–level approach to addressing this question. The first was to generate several silencing resistant constructs (two tagged and one untagged). Expression of these constructs appeared to further reduce differentiation rather than rescue the defect, but did so even in the context of siCONT, raising the possibility that the expressed protein (which is part of a 8 subunit complex) was exerting a dominant negative effect. In this regard, we noted from the literature a number of instances in which Cops3 was knocked down with either lentivirus or siRNA, and none of those sources reported overexpressing a silencing resistant version of Cops3 as a means to rescue (Banda et al., 2005; Huang et al., 2009; Pang et al., 2017; Peth et al., 2007; Wang et al., 2012; Yan et al., 2011; Yu et al., 2012). Along these lines, a recent systematic analysis comparing multiple strategies for reconstituting protein expression in RNAi experiments, demonstrated that while off target effects could be convincingly ruled out using RNAi to 11 different targets of mitochondrial creatine kinase 1, it is not always possible to rescue cells by reconstitution of protein expression (Datler and Grimm, 2013). Therefore, as an alternative means of ruling out off target effects, we used multiple oligos directed against different target sequences (see Author response image 2) to ensure that the observed differentiation and signaling changes were specific to the knockdown of Cops3 and not an off target effect of the siRNA. These data have been included in the text in the Results section and in Figure 2—figure supplement 1.

**Author response image 2. respfig2:** 

4) The authors perform several PLA assays to illustrate direct protein interactions in vivo. Unfortunately, PLA assays are prone to give false positive results and the data would be more convincing if another junctions protein that does not physically interact with Dsg1/DP was used as a negative control for the experiment. Along these lines, the signal distribution in the SIM and PLA experiments appear different. It would help if the authors could provide phase contrast or junctions marker staining to better illustrate cell–cell adhesion in the PLA data.

To provide additional controls for the PLA experiments, in addition to knockdown of individual components already included in the manuscript (Dsg1, Dp and Cops3) we paired Cops3 antibodies with antibodies against the adherens junction transmembrane protein E–cadherin (E–cad) and the cytoplasmic adherens junction component, Α–catenin (A–cat). As a positive control, Β–catenin (B–cat) was paired with both E–cad and A–cat antibodies. These experiments clearly show that while Cops3 interacts with desmosomal molecules Dsg1 and Dp, it does not interact with adherens junction components. In addition, we used the cell–cell junction protein Plakoglobin, Pg, as a fluorescence marker for cell–cell junctions to determine the extent to which PLA signals co–localize with cell–cell interfaces. These data show that while not all desmosome–Cops3 interactions are in close proximity to the Pg signal, a large percentage of PLA dots do appear to be localized at cell–cell interfaces (~60% for Dsg1/Cops3; ~67% for Dp/Cops3), although to a somewhat lesser extent than the positive control PLA pairings Α–Catenin/Β–Catenin (~82%) and Ecad/Β–catenin (~88%). These results are now included in the text in the Results section and in Figure 1 and Figure 1—figure supplement 4. These data are consistent with the hypothesis that cytoplasmic Cops3 interacts with desmosome molecules in non–membrane and membrane associated cytoplasmic particles and vesicles that we have previously identified in these cells (Harmon et al., 2013; Nekrasova et al., 2013; Patel et al., 2014), in addition to their localization at the cell surface.